# Korean Red Ginseng Saponins Play an Anti-Inflammatory Role by Targeting Caspase-11 Non-Canonical Inflammasome in Macrophages

**DOI:** 10.3390/ijms24021077

**Published:** 2023-01-05

**Authors:** Hui-Jin Cho, Eojin Kim, Young-Su Yi

**Affiliations:** Department of Life Sciences, Kyonggi University, Suwon 16227, Republic of Korea

**Keywords:** Korean red ginseng, saponin, inflammation, caspase-11, non-canonical inflammasome, macrophage

## Abstract

We previously reported that Korean red ginseng (KRG) exerts an anti-inflammatory role through inhibiting caspase-11 non-canonical inflammasome in macrophages; however, the components responsible for the anti-inflammatory role remained unclear. This study explored the anti-inflammatory activity of the KRG saponin fraction (KRGSF) in caspase-11 non-canonical inflammasome-activated macrophages. KRGSF inhibited pyroptosis, pro-inflammatory cytokine secretion, and inflammatory mediator production in caspase-11 non-canonical inflammasome-activated J774A.1 cells. A mechanism study revealed that KRGSF-induced anti-inflammatory action was mediated via suppressing the proteolytic activation of caspase-11 and gasdermin D (GSDMD) in caspase-11 non-canonical inflammasome-activated J774A.1 cells. Moreover, KRGSF increased the survival of lethal septic mice. Taken together, these results reveal KRGSF-mediated anti-inflammatory action with a novel mechanism, by inhibiting caspase-11 non-canonical inflammasome in macrophages.

## 1. Introduction

Inflammation is an immune mechanism for removing pathogens and harmful stimuli [1]. The inflammatory response consists of priming and triggering [2]. Priming is an inflammation-preparation process that increases inflammatory molecule production via activating transcription factors by various pathogen-associated molecular patterns, including Pam3CSK4, poly(I:C), and lipopolysaccharide (LPS), while triggering is an inflammation-activation process via stimulating inflammasomes [2,3]. Inflammasome is a cytosolic protein complex that triggers inflammatory responses and is classified into two major groups. Canonical inflammasomes, including nucleotide-binding and oligomerization domain-like receptor family and absent in melanoma 2 inflammasomes, were identified first and reported as crucial factors in inflammatory responses and various immunopathologies [4]. Recently, non-canonical inflammasomes, including mouse caspase-11 and human caspase-4 and -5 inflammasomes, were discovered. These are activated by direct sensing of intracellular LPS and are distinct from canonical inflammasomes, but also play an essential role in inflammatory responses and immunopathologies [5,6,7,8,9,10,11]. Inflammasomes are activated via recognizing pathogen- or damage-associated molecular patterns by intracellular pattern recognition receptors (PRRs) [12]. Despite unique ligands for different PRRs, inflammasome activation shares downstream signaling pathways. Inflammasome activation promotes the proteolysis of gasdermin D (GSDMD), and N-terminal GSDMD (N-GSDMD) goes to membranes to create GSDMD pores and induce pyroptosis, inflammatory cell death. Inflammasome activation also promotes the proteolysis of pro-caspase-1, and active caspase-1 causes the proteolytic maturation and release of the pro-inflammatory cytokines via GSDMD pores [12].

Korean ginseng (*Panax ginseng*) is an herbal medicine and can be fermented into Korean red ginseng (KRG) to improve its stability and biological constituents. Ginsenosides are the major saponins in KRG that exert various pharmacological [13] and anti-inflammatory actions [14,15]; however, most studies have focused on the priming process. Despite recent studies demonstrating the anti-inflammatory actions of ginsenosides in the triggering process, canonical inflammasomes have been the main target of ginsenosides, while the anti-inflammatory actions of ginsenosides by targeting the non-canonical inflammasomes have remained poorly understood. This study explored KRG saponin fraction (KRGSF)-mediated anti-inflammatory effects with diclofenac, an FDA-approved non-steroidal anti-inflammatory drug, in vitro and in vivo, through suppressing caspase-11 non-canonical inflammasome activation in macrophages and the animal model of lipopolysaccharide (LPS)-challenged lethal sepsis. In this way, the molecular mechanisms were uncovered.

## 2. Results

### 2.1. Suppressive Role of KRGSF on Caspase-11 Non-Canonical Inflammasome-Activated Pyroptosis in Macrophages

The cytotoxicity of KRG, KRGSF, and KRGNSF was compared in J774A.1 cells. Unlike KRG and KRGNSF, severe cytotoxicity of KRGSF was observed at 100 μg/mL in J774A.1 cells (Figure 1A). Next, the inhibitory effects of KRG, KRGSF, and KRGNSF on caspase-11 non-canonical inflammasome-activated pyroptosis were evaluated in J774A.1 cells. KRGSF showed a more potent suppressive action on caspase-11 non-canonical inflammasome-activated pyroptosis compared to KRG and KRGNSF up to 50 μg/mL, while an inhibitory effect of KRGSF on pyroptosis was not observed from 100 μg/mL in J774A.1 cells (Figure 1B). Therefore, 25 and 50 μg/mL of KRGSF were used in this study. The inhibitory role of KRGSF on pyroptosis was investigated and compared with that of diclofenac in J774A.1 cells. Pyroptotic cell death (Figure 1C) and pyroptosis-induced LDH release (Figure 1D) were significantly inhibited by KRGSF (25 and 50 μg/mL), but not by diclofenac. J774A.1 cells were also protected from pyroptosis by KRGSF (25 and 50 μg/mL) and diclofenac (Figure 1E). These results suggest that, unlike diclofenac, KRGSF has a suppressive effect on caspase-11 non-canonical inflammasome-activated pyroptosis in macrophages.

### 2.2. Suppressive Role of KRGSF on Caspase-11 Non-Canonical Inflammasome-Activated Production of Inflammatory Mediators in Macrophages

The suppressive role of KRGSF on the caspase-11 non-canonical inflammasome-activated production of inflammatory mediators was investigated in macrophages. KRGSF inhibited secretion and mRNA expression of IL-1β and showed a better inhibitory effect than diclofenac at 50 μg/mL in caspase-11 non-canonical inflammasome-activated J774A.1 cells (Figure 2A,B). Secretion and mRNA expression of IL-18 were also inhibited by KRGSF, but not by diclofenac in caspase-11 non-canonical inflammasome-activated J774A.1 cells (Figure 2C,D). Moreover, KRGSF and diclofenac inhibited NO production in both LPS-activated and caspase-11 non-canonical inflammasome-activated J774A.1 cells (Figure 2E,F). Both KRGSF and diclofenac decreased the mRNA expression of inducible NO synthase (iNOS) in caspase-11 non-canonical inflammasome-activated J774A.1 cells (Figure 2G). These results suggest that KRGSF has an inhibitory effect on the caspase-11 non-canonical inflammasome-activated production of inflammatory mediators in macrophages.

### 2.3. Mechanism of KRGSF-Suppressed Caspase-11 Non-Canonical Inflammasome Activation in Macrophages

We explored the underlying mechanisms of this inhibition in macrophages. KRGSF inhibited the proteolysis of caspase-11 (Figure 3A) and GSDMD (Figure 3B) in caspase-11 non-canonical inflammasome-activated J774A.1 cells. Diclofenac, however, showed no and marginal inhibitory effect on the proteolysis of caspase-11 (Figure 3A) and GSDMD (Figure 3B) in caspase-11 non-canonical inflammasome-activated J774A.1 cells, respectively. KRGSF did not inhibit the direct binding of LPS with caspase-11 (Figure 3C). These results suggest that KRGSF inhibits pyroptosis and inflammatory mediators by inhibiting proteolysis of both caspase-11 and GSDMD, but not by inhibiting the direct binding of LPS with caspase-11 in macrophages.

### 2.4. In Vivo Suppressive Role of KRGSF on Lethal Septic Shock in Mice

KRGSF-mediated in vivo anti-inflammatory effects were investigated in lethal septic mice. KRGSF increased the survival of mice from lethal septic shock at 150 mg/kg (*p* = 0.000025), but not at 200 mg/kg (*p* = 0.442580) (Figure 4B). KRGSF was well tolerated without acute toxicities, such as weight changes and abnormal behaviors, during the entire experimental period (Figure 4C).

## 3. Discussion

Investigation into the ginsenoside-mediated anti-inflammatory effects of KRG have mainly focused on the priming process and canonical inflammasomes of the triggering process [13,14,15]. Our previous study reported a new mechanism by which KRG exerts anti-inflammatory action and ameliorates lethal sepsis in mice through inhibiting caspase-11 non-canonical inflammasome in macrophages [16]; however, the anti-inflammatory constituents in KRG that inhibit caspase-11 non-canonical inflammasome remain unknown. The present study is the first to demonstrate that KRGSF is a key anti-inflammatory factor in caspase-11 noncanonical inflammasome activation during macrophage-mediated inflammatory responses.

Since ginsenosides are steroidal saponins that are cytotoxic at higher doses [17], pharmacologically effective but minimal cytotoxic doses of KRGSF in macrophages were first evaluated. KRGSF showed an anti-pyroptotic effect without severe cytotoxicity below 100 μg/mL in J774A.1 cells; therefore, 25 and 50 μg/mL of KRGSF were selected for this study. Pyroptosis is a principal consequence of inflammasome activation [12]. KRGSF exerted anti-pyroptotic effects in J774A.1 cells, but, interestingly, diclofenac, an FDA-approved NSAID, did not show any anti-pyroptotic effect, suggesting that KRGSF has a better anti-pyroptotic effect and different anti-inflammatory mechanism than diclofenac.

Another principal consequence of inflammasome activation is the secretion of pro-inflammatory cytokines and inflammatory mediators through the GSDMD pores of pyroptotic cells [12,16]. KRGSF decreased the expression and secretion of IL-1β and IL-18 in caspase-11 non-canonical inflammasome-activated J774A.1 cells, but diclofenac showed less and no inhibitory effect, respectively, indicating that KRGSF has better inhibitory potential in mRNA expression and secretion of pro-inflammatory cytokines in caspase-11 non-canonical inflammasome-activated macrophages with a mechanism distinct from that of diclofenac. Interestingly, KRGSF reduced NO production in both LPS-activated and caspase-11 non-canonical inflammasome-activated J774A.1 cells. NO production is a representative event in the priming step of the inflammatory response [18]. Our results showed that NO production could be induced in both the priming and triggering steps of inflammatory responses, which were dose-dependently inhibited by KRGSF in macrophages. iNOS is the enzyme responsible for NO production [18]. KRGSF decreases mRNA expression of iNOS in caspase-11 non-canonical inflammasome-activated J774A.1 cells, suggesting KRGSF-inhibited NO production by decreasing iNOS expression in caspase-11 non-canonical inflammasome-activated macrophages. Diclofenac also showed inhibitory effects on NO production and iNOS mRNA expression in both LPS-activated and caspase-11 non-canonical inflammasome-activated J774A.1 cells, suggesting that the mechanisms by which KRGSF and diclofenac inhibit the production of pro-inflammatory cytokines and inflammatory mediators may be different in macrophages.

Next, KRGSF-mediated anti-inflammatory mechanisms were demonstrated in caspase-11 non-canonical inflammasome-activated macrophages. Caspase-11 senses LPS by direct interaction to form LPS–caspase-11 complexes, which generate LPS–caspase-11 oligomers via CARD-CARD binding [5,6,7,8,9,10,11]. Caspase-11 non-canonical inflammasome is activated by its proteolysis [19], which subsequently promotes GSDMD proteolysis and GSDMD pore formation, leading to GSDMD pore-mediated pyroptosis in macrophages [5,6,7,8,9,10,11]. KRGSF showed a stronger inhibitory effect on the proteolysis of caspase-11 and GSDMD than diclofenac in caspase-11 non-canonical inflammasome-activated J774A.1 cells, providing evidence that KRGSF plays a better anti-inflammatory role in caspase-11 non-canonical inflammasome-activated macrophages by preventing the proteolytic activation of caspase-11 and GSDMD through a different mechanism from diclofenac. Caspase-11-mediated direct recognition of LPS is the first essential step that activates caspase-11 non-canonical inflammasome [5,6,7,8,9,10,11], but KRGSF did not inhibit this process, indicating that KRGSF inhibits the downstream proteolytic activation of caspase-11 and GSDMD, rather than inhibiting the upstream caspase-11-mediated recognition of LPS to suppress caspase-11 non-canonical inflammasome activation in macrophages. Interestingly, our previous study reported that KRG inhibits the LPS–caspase-11 interaction [16], providing evidence that other constituents, such as non-saponins in KRG, may inhibit the LPS–caspase-11 interaction; however, this requires further investigation.

Finally, KRGSF-mediated in vivo anti-inflammatory effects were examined in a mouse model of lethal sepsis [20,21]. KRGSF increased the survival of septic at a lower dose (150 mg/kg), which provides evidence that, similar to the in vitro cytotoxicity results (Figure 1A,B), a high dose of KRGSF (200 mg/kg) may also increase susceptibility to LPS-induced toxicity in vivo. No significant weight loss was observed in KRGSF-administered septic mice, indicating that KRGSF does not have acute toxicity or adverse effects at the tested doses during the time of the experimental period.

In this study, we demonstrated KRGSF-mediated anti-inflammatory effects both in vitro and in vivo by decreasing the activation of caspase-11 non-canonical inflammasome in macrophages and also explained a novel mechanism of KRGSF-inhibited signaling pathways activated by caspase-11 non-canonical inflammasome in macrophages, as depicted in Figure 5. Based on the results of this and previous studies [20], KRGSF displays anti-inflammatory activity by inhibiting both the priming as well as triggering processes of inflammatory responses, suggesting that KRGSF suppresses inflammatory responses and alleviates immunological disorders associated with inflammation by modulating multiple inflammatory pathways in macrophages. In addition, KRGSF showed much better anti-inflammatory effects than diclofenac in this study, providing strong and clear evidence that the mechanisms of KRGSF-mediated anti-inflammatory action are distinct from those of diclofenac, and that KRGSF may be effective and safe for NSAID-failure patients. Although KRGSF exhibited inhibitory activity against caspase-11 non-canonical inflammasome activation in macrophages, KRGSF, unlike KRG, did not inhibit LPS-caspase-11 interaction; therefore, the specific inhibitory factor of LPS–caspase-11 interaction in KRG needs to be further investigated. KRG non-saponins are critical constituents of KRG; therefore, future studies investigating the anti-inflammatory potential of KRG non-saponins in caspase-11 non-canonical inflammasome-activated inflammatory responses in macrophages are required. In conclusion, this is the first study reporting anti-inflammatory activity of KRGSF through suppressing caspase-11 non-canonical inflammasome activation during the triggering process of inflammatory responses in macrophages. It is also the first study to describe the potential of KRGSF as a safe and effective therapeutic agent with a unique mechanism distinct from that of the current NSAIDs in preventing and treating numerous inflammatory, autoimmune, and infectious diseases.

## 4. Materials and Methods

### 4.1. Materials

KRGSF and its constituents (Table 1) were provided by the Korea Ginseng Corporation (Daejon, Korea). HEK293 and J774A.1 cells were obtained from the Korean Cell Line Bank (Seoul, Korea). Roswell Park Memorial Institute (RPMI) 1640 medium, Dulbecco’s Modified Eagle’s Medium (DMEM), fetal bovine serum (FBS), penicillin, streptomycin, polyethylenimine, mouse IL-1β and IL-18 enzyme-linked immunosorbent assay (ELISA) kits, and streptavidin agarose beads were obtained from Thermo Fisher Scientific (Waltham, MA, USA). LPS (O111:B4), Pam3CSK4, and biotin-LPS were obtained from InvivoGen (San Diego, CA, USA). FuGENE HD was obtained from Promega (Madison, WI). Quanti-LDH PLUS Cytotoxicity Assay Kit was obtained from BIOMAX (Seoul, Korea). Bovine serum albumin (BSA) and Xpert protease inhibitor cocktail solutions were obtained from GenDEPOT (Katy, TX, USA). Sodium deoxycholate (SDC), 3-(4,5-dimethylthiazol-2-yl)-2,5-diphenyltetrazolium bromide (MTT), and sodium dodecyl sulfate (SDS) were obtained from HanLab (Seoul, Korea). pCMV-Flag-caspase-11 was obtained from Addgene (Watertown, MA, USA). Easy-BLUE reagent was obtained from iNtRON Biotechnology (Seongnam, Korea). M-MLV reverse transcriptase and the primers used for quantitative real-time polymerase chain reaction (qPCR) were obtained from Bioneer (Daejeon, Korea). qPCRBIO SyGreen Mix was obtained from PCR Biosystems (London, UK). GSDMD, caspase-11, Flag, and β-actin antibodies were obtained from Abcam (Cambridge, UK) and Santa Cruz Biotechnology (Santa Cruz, CA, USA). The enhanced chemiluminescence (ECL) reagent was obtained from ELPIS-Biotech (Daejeon, Korea).

### 4.2. Animal Husbandry

C57BL/6 mice (female, eight-weeks) were obtained from ORIENT BIO (Seongnam, Korea). Mice were fed pelleted diets and water ad libitum under a 12 h light/dark cycle. The study was approved by the Institutional Animal Care and Use Committee of Kyonggi University (approval number: 2022-009).

### 4.3. Cell Culture and Treatment

J774A.1 cells were maintained in RPMI 1640 medium with 10% heat-inactivated FBS, penicillin, and streptomycin. HEK293 cells were maintained in DMEM supplemented with 10% FBS, penicillin, and streptomycin. The cells were then incubated at 37 °C and 5% CO_2_. J774A.1 cells were pretreated with the indicated doses of KRGSF for 1 h, and Pam3CSK4 was directly added to the KGRSF-containing initial media. After 4 h of incubation, the cells were transfected with LPS for 24 h by directly adding the LPS-Fugene HD complex to the KRGSF/Pam3CSK4-containing media.

### 4.4. Cell Viability Assay

J774A.1 cells were treated with the indicated doses of KRG, KRGSF, and KRG non-saponin fraction (KRGNSF) for 24 h. J774A.1 cells pretreated with the indicated doses of KRGSF or diclofenac (50 μM) for 1 h were treated with Pam3CSK4 (1 μg/mL) for 4 h and transfected with LPS (2.5 μg/mL) for 24 h. Cell viability was examined using an MTT assay [22].

### 4.5. Pyroptosis Assay

J774A.1 cells pretreated with the indicated doses of KRG, KRGSF, or KRGNSF for 1 h were treated with Pam3CSK4 (1 μg/mL) for 4 h and transfected with LPS (2.5 μg/mL) for 24 h. J774A.1 cells pretreated with the indicated doses of KRGSF or diclofenac (50 μM) for 1 h were treated with Pam3CSK4 (1 μg/mL) for 4 h and transfected with LPS (2.5 μg/mL) for 24 h. Pyroptosis of the cells was examined by measuring lactate dehydrogenase (LDH) activity in the media.

### 4.6. ELISA

J774A.1 cells pretreated with the indicated doses of KRGSF or diclofenac (50 μM) for 1 h were treated with Pam3CSK4 (1 μg/mL) for 4 h and transfected with LPS (2.5 μg/mL) for 24 h. IL-1β and IL-18 in the media were quantified via ELISA.

### 4.7. Quantitative Real-Time Polymerase Chain Reaction (qPCR)

Total RNA was isolated from the J774A.1 cells pretreated with the indicated doses of KRGSF or diclofenac (50 μM) for 1 h, followed by treatment with Pam3CSK4 (1 μg/mL) for 4 h, and transfection with LPS (2.5 μg/mL) for 24 h. cDNA was synthesized from total RNA and analyzed via qPCR. Primer sequences for qPCR are summarized in Table 2.

### 4.8. NO Production Assay

J774A.1 cells pretreated with the indicated doses of KRGSF or diclofenac (50 μM) for 1 h were treated with Pam3CSK4 (1 μg/mL) for 4 h, and transfected with LPS (2.5 μg/mL) for 24 h. NO production in the culture media was examined using a Griess assay [23].

### 4.9. Immunoblot Analysis

J774A.1 cells pretreated with the indicated doses of KRGSF or diclofenac (50 μM) for 1 h were treated with Pam3CSK4 (1 μg/mL) for 4 h, and transfected with LPS (2.5 μg/mL) for 24 h. Whole-cell lysates were prepared by lysing the cells in RIPA buffer on ice for 1 h. Whole-cell lysates and culture media were subjected to SDS polyacrylamide gel electrophoresis, transferred onto PVDF membranes, and subsequently blocked with 5% BSA. The membranes were incubated with primary antibodies and horseradish peroxidase-linked secondary antibodies, and the target proteins were detected by an ECL system.

### 4.10. Caspase-11 and LPS Binding Inhibition Assay

HEK293 cells transfected with pCMV-flag-caspase-11 were lysed in a binding buffer (50 mM Tris-HCl, pH 7.6, 150 mM NaCl, 2 mM EDTA, 1% Triton X-100, and protease inhibitor cocktail) on ice for 1 h, followed by incubation with the indicated amounts of KRGSF for 1 h at 4 °C with constant rotation. Biotin-conjugated LPS (1 μg) was immobilized onto streptavidin agarose beads (8 μL), and unconjugated ligands were removed by washing the resins thrice with the binding buffer. LPS-bound beads were then incubated with the lysates for 1 h at 4 °C with constant rotation and washed thrice with the binding buffer. Proteins were eluted using sample buffer and subjected to immunoblot analysis.

### 4.11. In Vivo Sepsis Model

C57BL/6 mice (*n* = 6/group) were administered to orally with the indicated doses of KRGSF five times every 12 h. Twelve hours after the final administration, C57BL/6J mice were intraperitoneally injected with LPS (30 mg/kg). The survival as well as body weight of these mice were determined for 40 h (Figure 4A).

### 4.12. Statistical Analysis

The results of three to five independent experiments are described as mean ± standard deviation, and the Mann–Whitney U test was used to determine statistical significance. *p* values < 0.05 were considered to be significant (* *p* < 0.05 and ** *p* < 0.01).

## Figures and Tables

**Figure 1 ijms-24-01077-f001:**
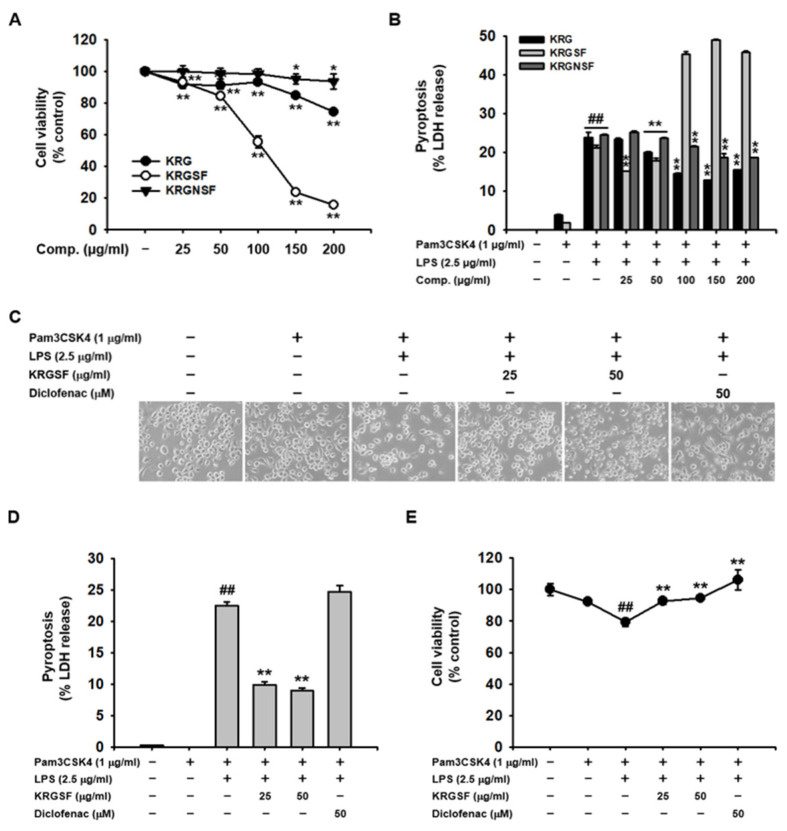
Suppressive role of KRGSF on caspase-11 non-canonical inflammasome-activated pyroptosis in J774A.1 cells. (**A**) J774A.1 cells were treated with the indicated doses of KRG, KRGSF, and KRGNSF for 24 h, and cell viability was examined using an MTT assay. (**B**) J774A.1 cells pretreated with the indicated doses of KRG, KRGSF, and KRGNSF for 1 h were treated with Pam3CSK4 (1 μg/mL) for 4 h, transfected with lipopolysaccharide (LPS: 2.5 μg/mL) for 24 h, and pyroptosis was examined by LDH release in the media. J774A.1 cells pretreated with the indicated doses of KRGSF or diclofenac (50 μM) were treated with Pam3CSK4 (1 μg/mL) for 4 h and transfected with LPS (2.5 μg/mL) for 24 h. (**C**) Cell morphology was photographed (magnification: 100×). (**D**) LDH release in the culture media was determined. (**E**) Cell viability was examined via an MTT assay. (**A**) * *p* < 0.05, ** *p* < 0.01 compared to the vehicle-treated negative control. (**B**,**D**,**E**) ^##^ *p* < 0.01 compared to the Pam3CSK4-treated control; ** *p* < 0.01 compared to the LPS-transfected positive controls.

**Figure 2 ijms-24-01077-f002:**
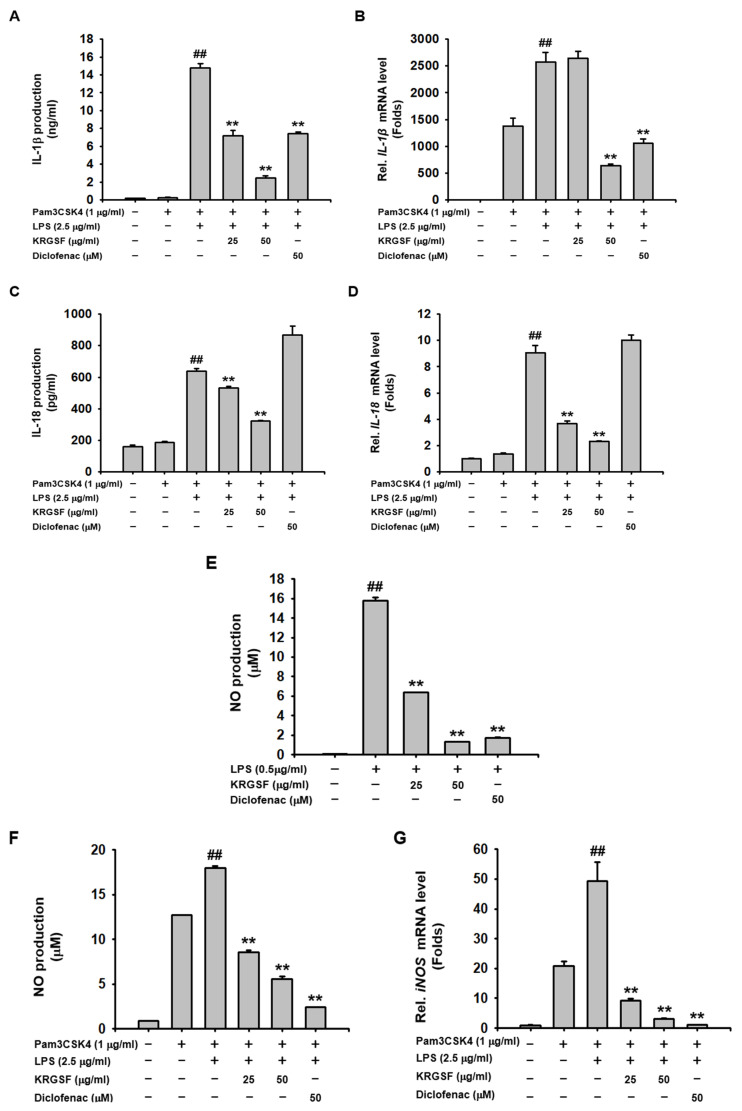
Suppressive role of KRGSF on caspase-11 non-canonical inflammasome-activated production of inflammatory mediators in J774A.1 cells. J774A.1 cells pretreated with the indicated doses of KRGSF or diclofenac (50 μM) for 1 h were treated with Pam3CSK4 (1 μg/mL) for 4 h and transfected with LPS (2.5 μg/mL) for 24 h. (**A**) IL-1β and (**C**) IL-18 in the media were examined by ELISA. mRNA of (**B**) IL-1β, (**D**) IL-18, and (**G**) iNOS were quantified by qPCR. NO in the media of J774A.1 cells pretreated with the indicated doses of KRGSF or diclofenac (50 μM) for 1 h, treated with (**E**) LPS for 24 h or (**F**) Pam3CSK4 (1 μg/mL) for 4 h, followed by transfection with LPS (2.5 μg/mL) for 24 h were examined by a Griess assay. (**A**–**D**,**F**,**G**) ^##^ *p* < 0.01 compared to the Pam3CSK4-treated control; ** *p* < 0.01 compared to the LPS-transfected positive controls. (**E**) ^##^
*p* < 0.01 compared to the vehicle-treated negative control; ** *p* < 0.01 compared to the LPS-treated control.

**Figure 3 ijms-24-01077-f003:**
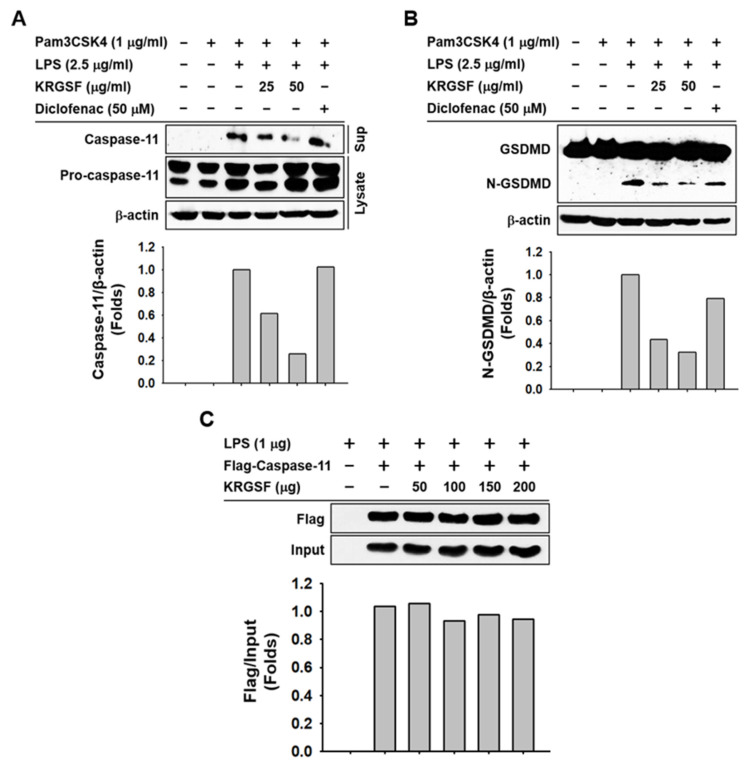
Mechanism of KRGSF-suppressed caspase-11 non-canonical inflammasome activation in J774A.1 cells. J774A.1 cells pretreated with the indicated doses of KRGSF for 1 h were treated with Pam3CSK4 (1 μg/mL) for 4 h and transfected with LPS (2.5 μg/mL) for 24 h. (**A**) Pro-caspase-11 and caspase-11 and (**B**) GSDMD and N-GSDMD in the whole-cell lysates and media were examined by immunoblot analysis. (**C**) Whole-cell lysates of the pCMV-flag-caspase-11-transfected HEK293 cells were reacted with biotin-LPS (1 μg) in the absence or presence of the indicated amounts of KRGSF for 1 h, followed by reaction with streptavidin agarose resins for another 1 h. Caspase-11 levels were determined via western blotting analysis. Band intensity was measured and plotted using the ImageJ program.

**Figure 4 ijms-24-01077-f004:**
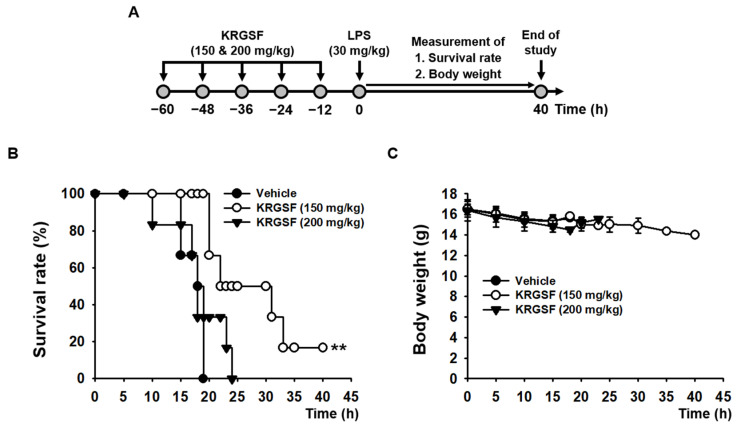
In vivo suppressive role of KRGSF on lethal septic shock in mice. (**A**) Experimental schedule. Mice orally administered KRGSF (150 and 200 mg/kg) were intraperitoneally injected with LPS (30 mg/kg). (**B**) Survival and (**C**) body weight were determined for 40 h. ** *p* < 0.01 compared to the vehicle-administered negative controls.

**Figure 5 ijms-24-01077-f005:**
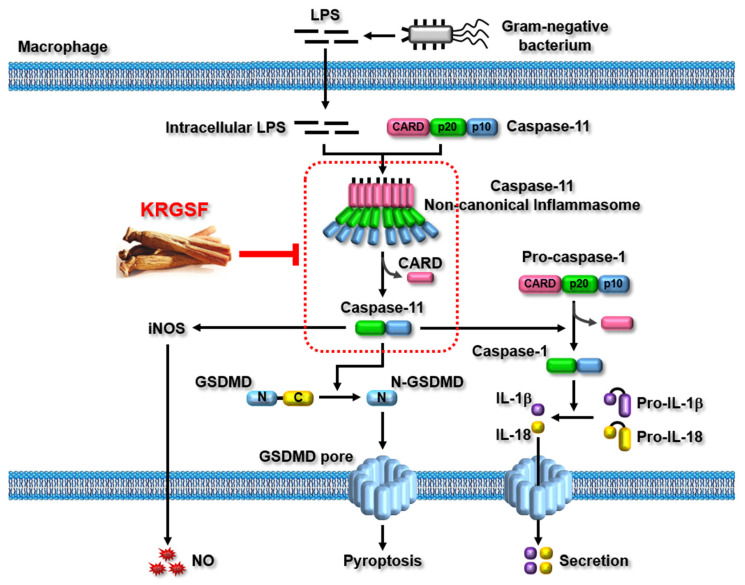
KRGSF-suppressed anti-inflammatory role in the activation of caspase-11 non-canonical inflammasome in macrophages.

**Table 1 ijms-24-01077-t001:** Composition of Korean Red ginseng saponin fraction (KRGSF) used in this study.

Saponins (Ginsenosides)	Amounts (mg/g)	Amounts (%)
Ginsenoside Rb1	108.15	28.26
Ginsenoside Rc	44.84	11.72
Ginsenoside Rb2	40.60	10.61
Ginsenoside Rg3s	34.14	8.92
Ginsenoside Re	29.60	7.74
Ginsenoside Rg1	24.72	6.46
Ginsenoside Rg2s	23.07	6.03
Ginsenoside Rh1	21.42	5.60
Ginsenoside Rf	21.42	5.60
Ginsenoside Rd	20.08	5.25
Ginsenoside Rg3r	14.61	3.82
**Total**	**382.65**	**100**

**Table 2 ijms-24-01077-t002:** Primer sequences used for PCR.

Target		Sequence (5′ to 3′)
*IL-1β*	For	GTGAAATGCCACCTTTTGACAGTG
	Rev	CCTGCCTGAAGCTCTTGTTG
*IL-18*	For	CAGCCTGTGTTCGAGGATATG
	Rev	TCACAGCCAGTCCTCTTACT
*iNOS*	For	GCCACCAACAATGGCAACAT
	Rev	TCGATGCACAACTGGGTGAA
*GAPDH*	For	CAATGAATACGGCTACAGCAAC
	Rev	AGGGAGATGCTCAGTGTT

## Data Availability

Not applicable.

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
