# Peer review of "Korean Red Ginseng Saponins Play an Anti-Inflammatory Role by Targeting Caspase-11 Non-Canonical Inflammasome in Macrophages"

_ijms, 2023, doi:10.3390/ijms24021077_

Round 1
Reviewer 1 Report
Cho et al. shows KRGSF inhibits pyroptosis, proinflammatory cytokine production and this is linked with suppression of proteolytic activation of Caspase11 and GasderminD in J774.1 macrophage cell line. Moreover, KRGSF prevents mice from septic shock with unknown reasons. However, rationale and connecting links between experiments and conclusions are missing, authors should use the reason connecting each experiment which will make more sense to the readers. Another point is that authors overestimated their findings without experimental evidence. In the introduction section, briefly mentions about Pam3CSK4, diclophenac, and LPS why included in this study? What are the inducers of Caspase-11 and when and how this is activated. Another drawback of this study is the only use of J774.1. Authors should repeat some of the important experiments in primary macrophage BMDMs which is easy to prepare. The in-vivo study is important and has clinical implications, however I have several concerns to be addressed-
Comments:
1. Abstract- KRGSF mediated proinflammatory cytokine inhibition at protein and mRNA level don’t require separate sentence.
2. Do authors know each gingenosides has similar function and dose match?
3. Why authors used only female mice in this study? Male mice show different pattern than the female?
4. LPS responds very quickly in macrophages, why authors went for 24h?
5. Line 160-162, authors linked LDH release with Caspase-11 whereas there is no experimental evidence here (I think reframe the sentence and use the word caspase-11 may linked and it was further validated). Pam3CSK4 is a TLR2 ligand and a poor inducer of Caspase 11 than the LPS itself. Importantly these two are not only limited to Caspase-11.
6. To be sure of caspase-11 involvement, use Caspase-11 siRNA and confirm its direct involvement.
7. Use fluorescence microscopy of flow to show pyroptotic cell death.
8. Fig 1D and 1E, authors used Pam3CSF4 only and with combinations of treatment. why authors don’t used only LPS as well. It is important to know and compare with LPS alone. Why is it necessary to use Pam3CSF4 and LPS together?
9. Fig 1 title: do authors think this is correct “Suppressive role of KRGSF on caspase-11 non-canonical inflammasome-activated pyroptosis in J774A.1 cells”. This is vage and it should be changed with matched finding of fig1.
10. Figure 2E and 2F are separately required?
11. Can authors provide raw data for cytokine and qPCR?
12. Fig 3A western blot is not convincing, to me it seems like a protein is unevenly transferred or film development defect. Please provide another western blot image.
13. Figure 3C- western blot don’t suggest LPS binding with Caspase-11, rather it suggests LPS activated Caspase-11 is unaffected by KRGSF treatment. LPS is well known to interact and oligomerize caspases, but it doesn’t suit here.
14. Fig 4B- there is no survival after 20 h LPS treatment (30 mg/kg) which is surprising to me, and I am more concerned if used animals were healthy at all?
Suggestion to author:
measure NF-kB activity assay and it could be a possible mechanism linked with deregulated proinflammatory cytokine and cell death.
Rewrite manuscript and elaborate findings with appropriate citations.
Methods:
1. Cells treatment is very confusing- authors added three separate treatments: first KRG/KRGSF/non-saponin fraction (these compounds are water soluble or in organic solvent), after one hr treatment supernatant was completely removed and washed with medium or PBS, or authors directly add Pam3CSF4 and LPS on top of the initially used medium. Make this clear in the method section.
2. Cell viability and pyroptosis assay section is again confusing with the treatment, write clearly exactly how it was followed.
3. Authors don’t think 2.5 ug/ml LPS is very high for macrophages? And LPS was treated or transfected?
4. In caspase-11 and LPS binding inhibition assay- after how long plasmid transfection this assay was performed? Authors tried this experiment in macrophages.
5. Griess assay and cytokine measurement what was the negative and background control? Authors used any positive or negative control for comparison. This should be mentioned in method section.
Author Response
Response to reviewer 1’s comments:
Cho et al. shows KRGSF inhibits pyroptosis, proinflammatory cytokine production and this is linked with suppression of proteolytic activation of Caspase11 and Gasdermin D in J774.1 macrophage cell line. Moreover, KRGSF prevents mice from septic shock with unknown reasons. However, rationale and connecting links between experiments and conclusions are missing, authors should use the reason connecting each experiment which will make more sense to the readers. Another point is that authors overestimated their findings without experimental evidence. In the introduction section, briefly mentions about Pam3CSK4, diclophenac, and LPS why included in this study?
- Brief explanation of Pam3CSK4, poly(I:C), LPS, and diclofenac have been added in the Introduction section.
What are the inducers of Caspase-11 and when and how this is activated.
- LPS is the only ligand that has been identified as an activator of caspase-11 non-canonical inflammasome. This was added in the Introduction section.
Another drawback of this study is the only use of J774.1. Authors should repeat some of the important experiments in primary macrophage BMDMs which is easy to prepare.
- Thank you for your suggestion, and I totally agree with your comment. We initially started this study using two murine macrophage cell lines, J774A.1 and RAW264.7, but RAW264.7 cells do not express ASC, a key adaptor molecule to activate canonical inflammasomes and to induce secretion of pro-inflammatory cytokines. Therefore, we instead used J774A.1 cells in this study. Also, we are currently setting-up the experimental protocol to prepare the primary macrophages, BMDMs, in our group. Therefore, we will definitely use both primary BMDMs and macrophage cell lines for our future studies.
The in-vivo study is important and has clinical implications, however I have several concerns to be addressed.
Comments:
1. Abstract-KRGSF mediated proinflammatory cytokine inhibition at protein and mRNA level don’t require separate sentence.
- Thank you for your comment. The sentence in the Abstract, ‘KRGSF also decreased gene expression of pro-inflammatory cytokines and an inflammatory enzyme in caspase-11 non-canonical inflammasome-activated J774A.1 cells’ has been deleted.
2. Do authors know each gingenosides has similar function and dose match?
- Although a number of studies have demonstrated the anti-inflammatory role of each ginsenoside, these studies have mostly focused on the priming step of inflammatory responses and the canonical inflammasome activation, and the studies demonstrating the role of ginsenoside in caspase-11 non-canonical inflammasome activation has been very limited. Therefore, the function of each ginsenoside in KRGSF used in this study (table 1) in caspase-11 non-canonical inflammasom-activated inflammatory responses is still unknown and needs to be investigated in the future.
3. Why authors used only female mice in this study? Male mice show different pattern than the female?
- Before this study, we examined the difference between male and female mice, but no difference in the LPS-stimulated lethal sepsis was shown in male and female mice. Also, many previous studies used female mice for LPS-stimulated lethal sepsis study (see below references). Theefore, we used female mice for this study.
1) Li et al., Shigella evades pyroptosis by arginine ADP-riboxanation of caspase-11. Nature. 2021 Nov;599(7884):290-295.
2) Kayagaki et al., Caspase-11 cleaves gasdermin D for non-canonical inflammasome signaling. Nature. 2015 Oct 29;526(7575):666-71.
3) Kayagaki et al., Noncanonical inflammasome activation by intracellular LPS independent of TLR4. Science. 2013 Sep 13;341(6151):1246-9.
4) Kayagaki et al., Non-canonical inflammasome activation targets caspase-11. Nature. 2011 Oct 16;479(7371):117-21.
4. LPS responds very quickly in macrophages, why authors went for 24h?
- In this study, the cells were treated with Pam3CSK4 (priming) and transfected (not treated) with LPS (triggering) to induce the activation of caspase-11 non-canonical inflammasome. Pam3CSK4 priming needs only 4 h-treatment, but LPS triggering needs 18 – 24 h transfection. In summary, for activating caspase-11 non-canonical inflammasome in macrophages, the cells should be treated with Pam3CSK4 for 4h, followed by transfection with LPS for 18 – 24 h. Therefore, the cells transfected with LPS for 24 h in this study. The references for this methodology are listed as bellows;
1) Kayagaki et al., Caspase-11 cleaves gasdermin D for non-canonical inflammasome signaling. Nature. 2015 Oct 29;526(7575):666-71.
2) Kayagaki et al., Noncanonical inflammasome activation by intracellular LPS independent of TLR4. Science. 2013 Sep 13;341(6151):1246-9.
3) Kayagaki et al., Non-canonical inflammasome activation targets caspase-11. Nature. 2011 Oct 16;479(7371):117-21.
4) Ye et al., Scutellarin inhibits caspase-11 activation and pyroptosis in macrophages. Acta Pharm Sin B. 2021 Jan;11(1):112-126.
5. Line 160-162, authors linked LDH release with Caspase-11 whereas there is no experimental evidence here (I think reframe the sentence and use the word caspase-11 may linked and it was further validated). Pam3CSK4 is a TLR2 ligand and a poor inducer of Caspase 11 than the LPS itself. Importantly these two are not only limited to Caspase-11.
- It is well known that the activation of caspase-11 non-canonical inflammasome induces GSDMD-mediated pyroptosis, and the most famous and well-established assay to determine pyroptosis is measuring LDH released in the cell culture media. Also, as described in comment 4, to activate the caspase-11 non-canonical inflammasome in macrophages, the cells are treated with Pam3CSK4 for 4 h, followed by LPS transfection for 24 h. The reason why the Pam3CSK4 is used for priming instead of LPS is to distinguish the effect extracellular effect of LPS (treatment) from the intracellular effect of LPS (transfection) in this experimental system. Pam3CSK4 priming and LPS transfection are very well-established method used for long time in many previous studies, and the references are as bellows;
1) Kayagaki et al., Caspase-11 cleaves gasdermin D for non-canonical inflammasome signaling. Nature. 2015 Oct 29;526(7575):666-71.
2) Kayagaki et al., Noncanonical inflammasome activation by intracellular LPS independent of TLR4. Science. 2013 Sep 13;341(6151):1246-9.
3) Kayagaki et al., Non-canonical inflammasome activation targets caspase-11. Nature. 2011 Oct 16;479(7371):117-21.
4) Ye et al., Scutellarin inhibits caspase-11 activation and pyroptosis in macrophages. Acta Pharm Sin B. 2021 Jan;11(1):112-126.
5) Lee et al., Caspase-11 auto-proteolysis is crucial for noncanonical inflammasome activation. J Exp Med. 2018 Sep 3;215(9):2279-2288.
6. To be sure of caspase-11 involvement, use Caspase-11 siRNA and confirm its direct involvement.
- It is a very good point. We are currently generating caspase-11 KO macrophage cell line and will examine your suggestion in future studies.
7. Use fluorescence microscopy of flow to show pyroptotic cell death.
- Thank you for your comment. Observing the pyroptotic cell death using fluorescence microscopy of flow will support our LDH results in this study. As described in comment 4 and 5, the primary and most well-established assay to determine pyroptosis is measuring LDH released in the cell culture media, that’s why we determined pyroptotic cell death by LDH assay. Since we currently do not have fluorescence microscopy in our institute, we will determine the pyroptotic cell death by both LDH assay and fluorescence microscopy in future studies.
8. Fig 1D and 1E, authors used Pam3CSF4 only and with combinations of treatment. why authors don’t used only LPS as well. It is important to know and compare with LPS alone. Why is it necessary to use Pam3CSF4 and LPS together?
- As described in comment 4 and 5, to activate the caspase-11 non-canonical inflammasome in macrophages, the cells should be ‘treated’ with Pam3CSK4 for 4 h and ‘transfected’ with LPS for 24 h, not treated with both Pam3CSK4 and LPS. It is well-established method from many previous studies.
9. Fig 1 title: do authors think this is correct “Suppressive role of KRGSF on caspase-11 non-canonical inflammasome-activated pyroptosis in J774A.1 cells”. This is vage and it should be changed with matched finding of fig1.
- The cytotoxicity of KRGSF was first evaluated in Fig. 1A to find non-toxic, but effective doses for this study. Fig. 1B compared the suppressive role of KRG, KRGSF, and KRGNSF in caspase-11 non-canonical inflammasome-activated pyroptosis in J774A.1 cells, and we found that 25 and 50 ug/ml of KRGSF were non-toxic, but effective in caspase-11 non-canonical inflammasome-activated pyroptosis in J774A.1 cells. Suppressive role of KRGSF on caspase-11 non-canonical inflammasome-activated pyroptosis were further evaluated in J774A.1 cells by observing cell shape (Fig. 1C), determining LDH release (Fig. 1D), and measuring cytotoxicity (Fig. 1D). These all results support that KRGSF suppresses caspase-11 non-canonical inflammasome-activated pyroptosis. Therefore, we are sure that the title of Fig.1 “KRGSF suppressed the caspase-11 non-canonical inflammasome-activated pyroptosis in J774A.1 cells” is correct based on the study results.
10. Figure 2E and 2F are separately required?
- Fig. 2E shows the inhibitory effect of KRGSF on NO production in the “LPS-priming (treatment) macrophages”, but Fig. 2F shows the inhibitory effect of KRGSF on NO production in the “Pam3CSK4 priming and LPS-transfecting macrophages where caspase-11 non-canonical inflammasome is activated”. Although these two experiments examined NO production in macrophages, the in vitro systems the meaning of the results are totally different, that’s why Fig. 2E and 2F were separated.
11. Can authors provide raw data for cytokine and qPCR?
- Sure, the raw data for cytokine ELISA and qPCR are described as bellows;
1) IL-1b ELISA raw data
|
Pam3CSK4 (1ug/ml) |
- |
+ |
+ |
+ |
+ |
+ |
|
LPS (2.5ug/ml) |
- |
- |
+ |
+ |
+ |
+ |
|
KRGSF (ug/ml) |
- |
- |
- |
25 |
50 |
- |
|
Diclofenac (uM) |
- |
- |
- |
- |
- |
50 |
|
AVE |
0 |
238 |
26650 |
14949 |
3321 |
17713 |
|
SD (pg/ml) |
0 |
3.913754 |
613.5899 |
17.65038 |
84.56983 |
280.0787 |
2) IL-1b Real time-PCR raw data
|
Pam3CSK4 (1ug/ml) |
- |
+ |
+ |
+ |
+ |
+ |
|
LPS (2.5ug/ml) |
- |
- |
+ |
+ |
+ |
+ |
|
KRGSF (ug/ml) |
- |
- |
- |
25 |
50 |
- |
|
Diclofenac (uM) |
- |
- |
- |
- |
- |
50 |
|
Expression |
1.0000 |
1375.8813 |
2577.2424 |
2642.0230 |
635.9075 |
1061.2196 |
|
SD |
0.2723 |
148.6797 |
176.1434 |
128.1146 |
31.8934 |
75.9860 |
3) IL-18 ELISA raw data
|
Pam3CSK4 (1ug/ml) |
- |
+ |
+ |
+ |
+ |
+ |
|
LPS (2.5ug/ml) |
- |
- |
+ |
+ |
+ |
+ |
|
KRGSF (ug/ml) |
- |
- |
- |
25 |
50 |
- |
|
Diclofenac (uM) |
- |
- |
- |
- |
- |
50 |
|
AVE |
160.6 |
186.5 |
637.8 |
532.4 |
321.35 |
867.5 |
|
SD (pg/ml) |
9.047099 |
6.522432 |
15.51292 |
8.644721 |
5.04262 |
55.93707 |
4) IL-18 Real time-PCR raw data
|
Pam3CSK4 (1ug/ml) |
- |
+ |
+ |
+ |
+ |
+ |
|
LPS (2.5ug/ml) |
- |
- |
+ |
+ |
+ |
+ |
|
KRGSF (ug/ml) |
- |
- |
- |
25 |
50 |
- |
|
Diclofenac (uM) |
- |
- |
- |
- |
- |
50 |
|
Expression |
1.0000 |
1.37401 |
9.04691 |
3.65140 |
2.31973 |
9.99742 |
|
SD |
0.04274 |
0.05048 |
0.55156 |
0.20567 |
0.05218 |
0.42284 |
12. Fig 3A western blot is not convincing, to me it seems like a protein is unevenly transferred or film development defect. Please provide another western blot image.
- Thank you for your comment. This blot is caspase-11 in the cell culture media, and we previously tried several times to detect the clean caspase-11 bands, but it was difficult to detect clean bands in the cell culture media, and this is the best images we get. We need to improve to detect the clean protein bands in the cell culture media, and please understand our current experimental situation.
13. Figure 3C- western blot don’t suggest LPS binding with Caspase-11, rather it suggests LPS activated Caspase-11 is unaffected by KRGSF treatment. LPS is well known to interact and oligomerize caspases, but it doesn’t suit here.
- This is a well-established in vitro binding assay to examine the direct interaction between LPS and caspase-11. As described in Method 2.10, Flag-Caspase-11 expressed in HEK293 cells was directly incubated with LPS-biotin in the absence or presence of KRGSF, and Flag-Caspase-11 was detected by Western blot analysis after LPS pull-down using streptavidin. This is a popular method to determine the direct binding between LPS and caspase-11 that has been used in the previous studies (see below references).
1) Shi et al., Inflammatory caspases are innate immune receptors for intracellular LPS. Nature. 2014 Oct 9;514(7521):187-92.
2) Lee et al., Caspase-11 auto-proteolysis is crucial for noncanonical inflammasome activation. J Exp Med. 2018 Sep 3;215(9):2279-2288.
14. Fig 4B- there is no survival after 20 h LPS treatment (30 mg/kg) which is surprising to me, and I am more concerned if used animals were healthy at all?
- This is a caspase-11 non-canonical inflammasome-activated acute lethal sepsis model in mice, which is well-established. In many previous studies, although the death time depends on the injected LPS doses, the mice are all dead around 20 – 24 h after LPS injection (see below references). Fig. 4 shows that KRGSF protected the mice from acute lethal sepsis, and the survived mice (KRGSF group) showed no weight loss, abnormal behavior, and adverse effect during the entire period of experiments. However, pathological evaluation of the organs of the KRGSF-administered mice also needs to be examined.
1) Kayagaki et al., Non-canonical inflammasome activation targets caspase-11. Nature. 2011 Oct 16;479(7371):117-21.
2) Kayagaki et al., Noncanonical inflammasome activation by intracellular LPS independent of TLR4. Science. 2013 Sep 13;341(6151):1246-9.
3) Hager et al., Cytoplasmic LPS activates caspase-11--implications in TLR4-independent endotoxic shock. Science. 2013 Sep 13;341(6151):1250-3.
4) Lee et al., Caspase-11 auto-proteolysis is crucial for noncanonical inflammasome activation. J Exp Med. 2018 Sep 3;215(9):2279-2288.
Suggestion to author:
measure NF-kB activity assay and it could be a possible mechanism linked with deregulated proinflammatory cytokine and cell death.
- Thank you for your suggestion. NF-kB is a key molecule activated in the priming step of inflammatory responses (sensing of PAMPs by extracellular PRRs), but this study demonstrated the inhibitory role KRGSF in caspase-11 non-canonical inflammasome activation during the triggering step of inflammatory responses (sensing of PAMP, LPS by intracellular PRR, caspase-11). NF-kB is not a key molecule directly involved in the inflammasome-activated signaling pathways, and the key molecules directly involved in the caspase-11 non-canonical inflammasome-activated signaling pathways are GSDMD, caspase-11, and pro-inflammatory cytokines, IL-1b and IL-18. Therefore, we evaluated the inhibitory effect of KRGSF on the activation of these molecules instead of NF-kB in this study. However, the demonstrate the link between priming and triggering steps of inflammatory responses, determining the activities of NF-kB and MAPKs in inflammasome-activated inflammatory responses may be required.
Rewrite manuscript and elaborate findings with appropriate citations.
- Thank you for your suggestion. The manuscript has been rewritten according to your comments and suggestions. Also, the citations which are already included in the manuscript have been appropriately described in the response to your comments.
Methods:
1. Cells treatment is very confusing- authors added three separate treatments: first KRG/KRGSF/non-saponin fraction (these compounds are water soluble or in organic solvent), after one hr treatment supernatant was completely removed and washed with medium or PBS, or authors directly add Pam3CSF4 and LPS on top of the initially used medium. Make this clear in the method section.J774A.21
- J774A.1 cells were pretreated with the indicated doses of KRGSF for 1 h, and Pam3CSK4 was directly added to the KGRSF-containing initial media. After 4 h incubation, the cells were transfected with LPS for 24 h by directly adding the LPS-Fugene HD complex to the KRGSF/Pam3CSK4-containing media. This detailed method for cell treatment has been added in section 2.3. Cell culture and treatment.
2. Cell viability and pyroptosis assay section is again confusing with the treatment, write clearly exactly how it was followed.
- The detailed methods for cell treatment of these assays have been added in section 2.3. Cell culture and treatment.
3. Authors don’t think 2.5 ug/ml LPS is very high for macrophages? And LPS was treated or transfected?
- LPS was transfected, and to activate caspase-11 non-canonical inflammasome, generally, 2.5 ug/ml LPS is used. Please see references below;
1) Kayagaki et al., Non-canonical inflammasome activation targets caspase-11. Nature. 2011 Oct 16;479(7371):117-21.
2) Kayagaki et al., Noncanonical inflammasome activation by intracellular LPS independent of TLR4. Science. 2013 Sep 13;341(6151):1246-9.
3) Hager et al., Cytoplasmic LPS activates caspase-11--implications in TLR4-independent endotoxic shock. Science. 2013 Sep 13;341(6151):1250-3.
4) Ye et al., Scutellarin inhibits caspase-11 activation and pyroptosis in macrophages. Acta Pharm Sin B. 2021 Jan;11(1):112-126.
4. In caspase-11 and LPS binding inhibition assay- after how long plasmid transfection this assay was performed? Authors tried this experiment in macrophages.
- HEK293 cells were transfected with Flag-caspase-11 expression plasmids for 48 h. The whole lysates of these cells were incubated with LPS-conjugated Biotin for 1 h in the absence or presence of KRGSF. After washing three times with binding buffer, the LPS-caspase-11 complexes were pulldowned using streptavidin-conjugated beads. The beads were then washed with PBS three times and subjected to Western blot analysis using Flag antibody. As per your suggestion, this experiment might be required in macrophages after the activation of the caspase-11 non-canonical inflammasome, but before this experiment, we first tried to find the clue of the direct binding between LPS and caspase-11 in this study using a well-established method in the previous studies, as bellows;
1) Shi et al., Inflammatory caspases are innate immune receptors for intracellular LPS. Nature. 2014 Oct 9;514(7521):187-92.
2) Lee et al., Caspase-11 auto-proteolysis is crucial for noncanonical inflammasome activation. J Exp Med. 2018 Sep 3;215(9):2279-2288.
5. Griess assay and cytokine measurement what was the negative and background control? Authors used any positive or negative control for comparison. This should be mentioned in method section.
- Negative and background controls in this study are vehicle-treated groups. Positive controls are the groups that were treated with Pam3CSK4 and transfected with LPS (3rd group in the graphs). Negative and positive groups were described in the legends of each figure.

Reviewer 2 Report
Plant-based traditional medicine is adopted as a cultural practice which purely based on observation. Though the advancement of science discovered the potential molecules responsible for the therapeutic benefits of many plants, there is still a vastly unexplored area that opens opportunities for scientists.
This study by Cho JH et al., is a highly welcomed, well-designed, and presented study that investigated the mechanism behind the anti-inflammatory property of KRG saponin fraction.
Murine sepsis model utilization to confirm their hypothesis makes this manuscript a completion. I encourage authors to extend their field of application from sepsis models to cancer models.
Kudos to the authors.
Author Response
Plant-based traditional medicine is adopted as a cultural practice which purely based on observation. Though the advancement of science discovered the potential molecules responsible for the therapeutic benefits of many plants, there is still a vastly unexplored area that opens opportunities for scientists.
This study by Cho JH et al., is a highly welcomed, well-designed, and presented study that investigated the mechanism behind the anti-inflammatory property of KRG saponin fraction.
Murine sepsis model utilization to confirm their hypothesis makes this manuscript a completion. I encourage authors to extend their field of application from sepsis models to cancer models.
- We highly appreciate your comment and suggestions. Recent emerging studies are trying to demonstrate the role of canonical and also caspase-11 non-canonical inflammasomes in the pathogenesis of various types of cancers since chronic inflammation is a key risk factor for cancer development. Therefore, as per your suggestion, studying the roles of caspase-11 non-canonical inflammasome in cancer development and progression will be highly required.

Round 2
Reviewer 1 Report
Cho et al. modified the manuscript, however many to improve. I appreciate authors for justifying their findings and methodology with the previous reports, however, I encourage not to blindly follow previously published study, of course they are important to the field but this is the way how science improves.
I couldn't understand the author's inhibition for not to repeat or establishing primary macrophage culture to validate some of the important findings.
I agree with the importance of the finding specially the in-vivo results though it lacks mechanism.